# Prevalence of Microplastics in the Eastern Oyster *Crassostrea virginica* in the Chesapeake Bay: The Impact of Different Digestion Methods on Microplastic Properties

**DOI:** 10.3390/toxics10010029

**Published:** 2022-01-10

**Authors:** Thet Aung, Inayat Batish, Reza Ovissipour

**Affiliations:** 1Department of Food Science and Technology, Virginia Tech, Blacksburg, VA 24061, USA; theta@vt.edu (T.A.); inayatbatish@vt.edu (I.B.); 2FutureFoods Lab and Cellular Agriculture Initiative, Seafood Agricultural Research and Extension Center, Hampton, VA 23669, USA; 3Center for Coastal Studies, Virginia Tech, Blacksburg, VA 24060, USA

**Keywords:** microplastics, bivalves, isolation, Chesapeake Bay, Raman spectroscopy

## Abstract

This study aimed to determine the microplastic prevalence in eastern oysters (*C. virginica*) in three sites in the Chesapeake Bay in Virginia and optimize the digestion methods. The digestion results illustrate that the lowest recovery rate and digestion recovery were related to enzymatic, enzymatic + hydrogen peroxide (H_2_O_2_), and HCl 5% treatments, while the highest digestion recovery and recovery rate were observed in H_2_O_2_ and basic (KOH) treatments. Nitric acid digestion resulted in satisfying digestion recovery (100%), while no blue polyethylene microplastics were observed due to the poor recovery rate. In addition, nitric acid altered the color, changed the Raman spectrum intensity, and melted polypropylene (PP) and polyethylene terephthalate (PET). In order to determine the number of microplastics, 144 oysters with an approximately similar size and weight from three sites, including the James River, York River, and Eastern Shore, were evaluated. Fragments were the most abundant microplastics among the different microplastics, followed by fibers and beads, in the three sites. A significantly higher number of fragments were found in the James River, probably due to the greater amount of human activities. The number of microplastics per gram of oyster tissue was higher in the James River, with 7 MPs/g tissue, than in the York River and Eastern Shore, with 6.7 and 5.6 MPs/g tissue.

## 1. Introduction

Global plastic production has increased in recent decades from 1.9 tons in 1950 to 368 million tons in 2019 [1]. Approximately 269,000 tons of plastic is floating on the ocean surface, equivalent to 5.25 trillion plastic particles [2], with an abundance of 103 to 105 particles per m^3^ [3] or 0.001 to 0.1 particles per mL [4]. The current estimates of total plastic in the world’s oceans may be underestimated since 50% of the plastics are negatively buoyant and, as a result, may sink to the bottom of the ocean [2]. 

A wide range of plastic polymers, with different shapes (e.g., spheres, fiber, film, irregular) and different densities, have been detected in the ocean and marine organisms, comprising polyethylene (PE), polypropylene (PP), polyethylene terephthalate (PET), polyvinylchloride (PVC), polyester, polystyrene, and polyamide [5,6,7,8,9,10]. Different densities make microplastics (MPs) across the water column, from the surface to the bottom of the ocean, impact MP availability in marine organisms. Every marine organism tested to date has been shown to ingest MPs [11], and in many cases, translocation of MPs from the digestive tract to other organs has been reported [11].

When plastics enter the ocean, the degradation is dependent on the polymer composition, shape, and density of the plastics in combination with the environmental conditions such as weathering, temperature, irradiation, and pH [12]. Plastic particles contaminating the environment can negatively impact the ecosystem by influencing the food web, from microorganisms to marine mammals, easily bioaccumulated and transferred via food to humans [13]. Large plastic particles are often found in the digestive tracts of marine vertebrates or birds, while smaller particles such as MPs can be translocated to the circular system or into surrounding tissue [11]. Ingested MPs by more than 140 different species of marine organisms showed negative impacts, including DNA damage in clams [14], delayed larvae development in oysters [15], valve closure in mussels [16], transcriptional alternation in mussels [17], and reduced energy budget in crabs [18]. Among marine organisms, filter-feeding bivalves are exposed to MPs to a greater extent since they are sessile and non-selectively filter water, resulting in bioaccumulation of MPs, having a negative impact on their performance, reducing production yield in aquaculture and fisheries, negatively impacting rural and coastal communities, and possibly serving as a vehicle for the transfer of MPs to humans. 

Aquaculture and fisheries are rapidly growing industries, and their production has almost doubled in the last ten years, with 179 million tons of seafood produced in 2018 and additional increases in production anticipated in the future [19]. In the U.S., the annual per capita consumption of seafood products is around 19.2 kg and is projected to approach 22.5 kg by 2030 [19]. Oysters are the top marine aquaculture product, which justifies a thorough investigation of how MPs can negatively impact the oyster industry. Recent research showed that oysters could ingest MPs, and after a two-month exposure to polystyrene pellets, the oysters exhibited a significant decrease in sperm velocity and quantity, size of oocytes, and larvae development [15]. This study demonstrates the potential impact of MPs on oyster production and the importance of assessing MP prevalence in oysters. Multiple studies have examined the type and abundance of MPs in field-collected organisms [20,21,22,23]. However, to the best of our knowledge, there is only one study on the prevalence of MPs in oysters in Florida [23] and one study on oysters in Georgia [24]. In the Chesapeake Bay, there are only a few studies on MPs in the water [25,26]. Yonkos et al. [25] studied the abundance of MPs in water samples in four estuarine rivers in the Chesapeake Bay in Maryland, and Yanez et al. [26] evaluated the number of MPs in the water in the James River and York River. However, there is no report on the prevalence of MPs in oysters in the Chesapeake Bay. Considering that MPs might influence the oyster industry, more research is required for understanding the abundance of MPs.

Quantifying MPs in seafood requires an efficient isolation step to remove all the organic materials and soft tissues without affecting the polymer integrity [27]. Four different digestion methods have been used for eliminating organic materials including acids [21,28,29,30], bases [30,31,32,33], oxidative agents [30,34,35], and enzymes [36,37]. Most of these methods are corrosive, affecting the MP structure, expensive, and time consuming. For example, many researchers found that nitric acid can damage the polymers and, in many cases, melt them, resulting in underestimating MPs in marine organisms. Thus, this study aimed to evaluate the influence of different digestion methods on standard MP recovery and chemical structures. In addition, in this study, we determined the prevalence of MPs in oysters from three sites in the Chesapeake Bay in Virginia for the first time. 

## 2. Materials and Methods

### 2.1. Experiment 1: Optimizing the Digestion Method

#### 2.1.1. Selecting the Digestion Method

During the entire experiment, to prevent any cross-contact, all the solvents were filtered using 0.45 um filter paper, and all the glassware was washed with 1 M nitric acid and rinsed three times with deionized water, then washed with 70% ethanol and dried in an oven, and then covered with aluminum foil. Six digestion approaches were used for removing the organic tissue from 10 oysters for each approach to collect the blue polyethylene microplastics (300–355 µm) to determine the digestion efficiency and recovery rate. Approach 1 was selected based on enzymatic hydrolysis, using 3% Alcalase at 60 °C for 48 h. Approach 2 was based on Approach 1, followed by hydrogen peroxide 30% and incubation for 24 h at 60 °C. Approach 3 was based on using 30% hydrogen peroxide at the ratio of 40:1 and digesting the samples for 24 h at 65 °C, continued by 48 h digestion at room temperature, and adding 30 ppt sodium chloride and maintaining samples at room temperature for another 24 h. Approach 4 was selected based on the conventional digestion method using 69% nitric acid and incubation at 60 °C for 24 h. Approach 5 was based on using HCl 5% at 60 °C for 24 h. Approach 6 was selected based on using KOH 10% and incubation at 60 °C for 24 h. All the samples were filtered after digestion, using 0.45 filter paper via a vacuum filter. Filters were dried at room temperature under the hood for 3 h. Digestion efficiency (%) was calculated using the following equation according to Karami et al. [27]: Digestion efficiency (%) = (*Wi* − (*Wa* − *Wb*))/*Wi* × 100
where *Wi* is the initial weight of the biological materials, *Wa* is the weight of the dry filter membrane after filtration, and *Wb* is the weight of the dry filter membrane before filtration. Karami et al. [27] set more than 95% digestion efficiency as the acceptable threshold to reduce the optical examination of MPs by organic materials left from the digestion. 

To determine the proper digestion method with the highest recovery rate, 5 g of oyster soft tissue was mixed with blue polyethylene microplastics (300–355 µm, Cospheric LLC., Goleta, CA, USA) at a rate of 20 microplastics per gram of oyster tissue (total of 100 MPs per sample) in triplicates (*n* = 3). Oysters were obtained from local stores for experiment 1. The MP recovery rate was calculated based on the number of microplastics added to the raw materials and the number of MPs recovered after digestion and filtration. 

#### 2.1.2. The Impact of Digestion Methods on Plastic Chemical Structure 

Raman spectra of the intact plastics (control), including polypropylene (PP), polyethylene terephthalate (PET), and polystyrene (PS), before and after exposure to different digestion methods (Approaches 1–6), were collected in the range of 2500–500 cm cm^−1^ using a DXR2 microscopy Raman spectrometer (Thermo Fisher Scientific Inc., Waltham, MA, USA) equipped with a 785 nm diode laser source. 

### 2.2. Experiment 2: Microplastic Prevalence in Oysters from the Chesapeake Bay 

#### 2.2.1. Sampling Sites

Samples for this study were collected from the Chesapeake Bay area in Virginia from three different locations (Figure 1) with a salinity range of 10 to 25 ppt during the summer of 2020. We selected the Chesapeake Bay watershed due to the degraded stormwater runoff, rainfall, groundwater, and water from canals from nearby urban and suburban areas, as well as the sea level rise and high-tide flooding, which has critically polluted the watershed [38,39]. After collecting the samples from the sites, oysters were placed into coolers with ice and transferred to the lab within 2 h. 

#### 2.2.2. Sample Preparation 

Upon arrival in the lab, whole live oysters were placed in zip lock bags and were frozen for 24 h before processing. Oysters were thawed at room temperature for further studies, shell length and width were measured, and after removing the soft tissue, the weight was recorded before the digestion process. Before the digestion process, 500 mL Erlenmeyer flasks were washed three times with deionized water, previously filtered with 0.2 µm nitrocellulose membrane filter paper using vacuum filtration. Each oyster soft tissue was cut into small pieces and then placed into the 500 mL flask. Samples were digested using the chemical digestion method according to Li et al. [40] and the NOAA standard procedure for marine debris [41]. First, 200 µL of 0.1% Tween 20 and 30% hydrogen peroxide at the ratio of 40:1 (200 mL:5 g) was added. Then, flasks were incubated in a shaking incubator at 65 °C for 24 h, followed by maintaining them at room temperature for another 48 h. Then, 30 g sodium chloride was added to each flask, and flasks were kept at room temperature for another 24 h. The top solution was filtered using a 0.45 µm filter and vacuum filtration. 

In total, 144 live oyster samples were collected from three sites with no significant differences in soft tissue, shell length, and shell width (*p* < 0.05). Mean shell lengths for sites 1, 2, and 3 were 8.9, 9.2, and 9.3 cm, respectively. The mean weight of soft tissue for sites 1, 2, and 3 was 18.6, 19.4, and 20 g, respectively (Table 1). 

### 2.3. Data Analysis

The results are presented as the mean of the replicates ± standard deviation. The significance of the differences was determined with one-way analysis of variance (*p* < 0.05). Raman spectra were pre-processed by applying baseline correction, normalization, and smoothing to reduce the noise using Unscrambler^®^ X software (version 10.5) (CAMO Software, Oslo, Norway). Microplastic color and size were measured using a Nikon Eclipse Ci microscope (Melville, NY, USA), and ImageJ software. 

## 3. Results

### 3.1. Experiment I: Optimizing the Digestion Methods

This experiment was conducted to evaluate the impact of six digestion approaches on the digestion efficiency and recovery rate of standard blue polyethylene (300–355 µm), and the chemical structure of three plastics, including PP, PET, and PS (Table 2, Figure 2). The results illustrate that treatment with H_2_O_2_, KOH, and HNO3 had the highest digestion efficiency, meaning that the organic tissues were digested completely (*p* < 0.05). Meanwhile, enzymatic hydrolysis, enzymatic-H_2_O_2_, and HCl 5% treatments showed a low digestion efficiency. The recovery rate for the blue PE microplastics was satisfying for H_2_O_2_ and KOH, while for HNO_3_, due to its strong acidity, the blue PE microplastics were melted completely. Strong acidic and basic solutions can digest organic tissue by degrading proteins, carbohydrates, and fats [35]. However, lower concentrated acids such as 5% HCl are not suitable for digesting a high amount of organic materials. Neulle et al. [34] and Karami et al. [27] also reported poor digestibility of tissues using 5% and 20% HCl solutions, respectively. Previous studies also reported that digesting bivalves in 30% H_2_O_2_ at 60–65 °C for 24 h followed by incubation at room temperature for 48 h resulted in complete digestion of soft tissues [27,40,42]. Von Friesena et al. [37] developed an efficient enzymatic digestion method with a high recovery rate (87%) and digestion efficiency (97%) for bivalve tissue, which is in contrast to our findings. This could be explained by the fact that we used the Alcalase enzyme, a proteinase enzyme, while the other researchers used pancreatic enzymes consisting of amylase, lipase, and proteinase, which can digest the whole tissue [37]. We also observed that the enzyme, enzyme-H_2_O_2_, and 5% HCl treatments, which partially digested the oysters, resulted in foam formation and clogged the filters, reducing the microplastics’ recovery and characterization. We selected Alcalase based on the fact that oysters have a higher protein content, and low carbohydrate content. However, Alcalase alone did not digest the oyster tissue completely. 

We also applied six digestion approaches to three plastics, including PP, PET, and PS, and characterized them using morphological changes and chemical structure changes via Raman confocal microscopy (Table 2 and Figure 2). 

The results of the Raman spectra analysis of untreated and chemically treated plastic materials illustrate that the chemicals caused changes in the spectrum peaks (Figure 2). These changes in peak intensity and band shifting might be due to polymer molecular alteration because of the chemical digestion [35]. Peak intensities around 810, 844, 976, 1152, 1171, 1131, 1438, and 1460 cm^−1^ were reduced significantly after chemical treatments in PP. Thiele et al. [33] found that the PP Raman spectrum peak intensity was significantly reduced after chemical treatments. The peak intensity around 1460 cm^−1^, which is assigned to asymmetric bending of the CH_3_ group, was significantly reduced after exposing PP to the chemicals, particularly HNO_3_. Karami et al. [27] also found that the Raman peak intensities were reduced for PP treated with HNO_3_. For PET, the chemical treatments reduced the peak intensity for the ring C = C stretching at 1610 cm^−1^ and C = O stretching band around 1722 cm^−1^, which is in agreement with other studies [27,42,43]. This reduction might be related to the depolymerization of the polymer structure compared to the control group [44]. For PS, the peak intensity around 1004 cm^−1^, which is related to the ring breathing mode, was increased after exposure to the chemicals, which might be due to the degradation of the polymer structure and rearrangement and aggregation of the polymer chains [27]. In addition, in PS treated with HNO_3_, the peak at 1350 cm^−1^ was very sharp, and the intensity was higher than the control and treated samples, which is due to the CH deformation. 

### 3.2. Experiment II: Prevalence of MPs in Oyster Samples 

The number of microplastics was not significantly different among the tested sites (*p* > 0.05). The average number of microplastics in each oyster from the James River (Site 3) was 140 microplastics per oyster, followed by the York River (Site 2) with 128 microplastics per oyster. The Eastern Shore (Site 1) had the lowest number of microplastics (104 per oyster). The highest number of microplastics per gram of soft tissue was found for the James River (7 MPs/g tissue), followed by the York River (6.7 MPs/g tissue) and Eastern Shore (5.6 MPs/g tissue). In all sampling locations, the highest number of microplastics was related to fragments (80–88%), followed by fibers (9–12%) and beads (2–6%) (*p* < 0.05). The most common colors were black, white, and transparent (Table 3). MPs found in this study had different size ranges, including 110–2300 µm for microfibers, 14–1280 µm for fragments, and 6–282 µm for beads.

The results of this study indicate the abundance of microplastics in oyster samples which were collected from three sites in the Chesapeake Bay in Virginia. Our results show that fragment microplastics were the most abundant microplastics in the oysters from the Chesapeake Bay. This is in contrast to studies that have found microfibers are the most abundant plastics in oyster tissue [23,24,32,45]. However, few studies have reported that fragment microplastics are the main microplastics in oysters [40,46,47]. Different factors could explain this, including the difference in species, filtering rate, sampling location, filtration, and instrumentation [47]. For example, Scircle et al. [47] used fluorescence microscopy targeting larger microplastics, and Rochman et al. [32] applied µFT-IR analysis for microplastics in bivalves and found that the most abundant microplastics are fragments. It has been widely shown that bead and fragment particles are most likely removed from the digestive tract. Our results suggest that fragment microplastics can also accumulate at a high concentration in oysters, which is in agreement with other studies [40]. In another study, samples from three rivers (Stroubles Creek, Roanoke River, and James River) in Virginia indicated that the most abundant microplastics are fragments (80%), followed by beads (15%) and fibers (5%) [48]. 

The average number of microplastic particles detected in oysters from three sites in the Chesapeake Bay in Virginia, with an average soft tissue weight of 18.6 to 20 g, was between 104 to 140 particles per oyster (5.6–7 particles per gram of soft tissue), which was higher than the microplastic particles found in other bivalves. Oysters (*C. virginica*) collected from a Florida estuary had 16.5 microplastic particles per oyster [23]. Others also reported lower numbers of microplastic particles in bivalves: 2 microplastics per oyster (*C. gigas*) (0.47 particles/g) [28], 0.2–0.3 particles per gram in *C. gigas* [29], and 0.18 MPs per gram of oyster (*C. virginica*) in Georgia [24]. While most studies showed lower numbers of microplastics in other bivalves, 10–29 particles per gram of tissue in oyster (*C. gigas*) in Germany [49], 34–178 particles per blue mussel (*Mytilus edulis*) in Canada [50], 2.1–10.5 particles per gram of nine different bivalves from a fishery market in China [40], and 1.5–7.2 particles per gram of tissue in oysters from the Pearl River Estuary in China [51] have been reported, indicating higher numbers of microplastics in some regions. Oysters in our study were also larger compared to those in other studies and had larger gills and lip pulps, which may result in the possibility of taking up more MPs [52]. Among marine organisms, filter-feeding bivalves are exposed to MPs to a greater extent since they are sessile and non-selectively filter water, resulting in bioaccumulation of MPs, having a negative impact on their performance, reducing production yield in aquaculture and fisheries, negatively impacting rural and coastal communities, and possibly serving as a vehicle for the transfer of MPs to humans. Recent research has shown that oysters could ingest MPs, and after intake, microplastics can attach to the guts, gills, and tissues, reducing energy uptake and impairing muscle function and reproduction [14,15,53]. Microplastics may also sorb harmful contaminants that, once ingested or incorporated in tissues, are released into the organism [54]. In some studies, environmental MP concentrations have been directly correlated with microplastic burdens in coastal bivalves [42,50].

The knowledge about microplastic quantification and distribution in the Chesapeake Bay is very limited to only a few studies [25,26,55]. The current study evaluated the prevalence of microplastics in oysters in the Chesapeake Bay in Virginia, which is the third largest oyster producer in the U.S. MP pollution has been reported in areas with a higher urban density, more commercial fishing, higher industrial waste discharge, more sewer overflows, more wastewater treatment plants, and more shipping ports [25,26,47]. The abundance of MPs in the surface water of the James River was 700–9000 MPs/l, and in the York River, it was 1400 to 15000 MPs/L, mainly due to the wastewater treatment plants near both rivers [26]. The MP concentration is also influenced by wind, rain, and extreme meteorological conditions such as hurricanes and floods, resulting in MP transfer from terrestrial environments to the sea [20,25,56,57,58,59]. For example, Moore et al. [58] reported that the amount of surface plastic debris with less than a 4.75 mm diameter in California surface waters near the Los Angeles stormwater system was six times higher after a storm. Heavy rains increase the amount of plastic debris entering coastal regions, and strong winds during hurricanes result in wave action, creating vertical mixing within the water column which can resuspend plastics [25]. Yonkos et al. [25] also observed a microplastic peak in September after Hurricane Irene and Tropical Storm Lee in the Chesapeake Bay. More studies on the prevalence of microplastics in the water, oysters, clams, crabs, and sediments from different locations of the Chesapeake Bay during different seasons are required to provide a deep understanding of the MP patterns in this region. 

## 4. Conclusions

The presence of MPs in the water, sediments, and seafood is an emerging concern. To evaluate the prevalence of MPs in seafood products, employing a proper digestion method to remove organic materials without negatively influencing MP chemical and physical properties is critical. In the current study, we evaluated the impact of six different digestion approaches on the digestion efficiency and MP recovery rate using standard blue PE MPs and found that H_2_O_2_ and KOH had the highest digestion efficiency and MP recovery. Meanwhile, HNO_3_ digested 100% of the organic tissue, but it melted all the blue PE MPs. Enzymatic, enzymatic-H_2_O_2_, and 5% HCl treatments resulted in poor digestion and MP recovery. In addition, our results from exposing PP, PET, and PS to these six approaches indicate that all the treatments could alter the chemical structure of the polymers based on Raman spectroscopy. Due to the high cost of H_2_O_2_, and its negative impact on the plastic chemical structure, KOH would be a suitable option for digesting organic tissues. In addition, our results indicate high amounts of MPs in oysters collected from three locations in the Chesapeake Bay in Virginia. MP fragments were the most abundant particles, followed by fibers and beads. More studies are needed to determine the MP prevalence in the water, sediments, oysters, and other organisms in the Chesapeake Bay. The collection of more samples in different seasons is also required due to different water inflow patterns, hurricane seasons, and water runoff into the bay. 

## Figures and Tables

**Figure 1 toxics-10-00029-f001:**
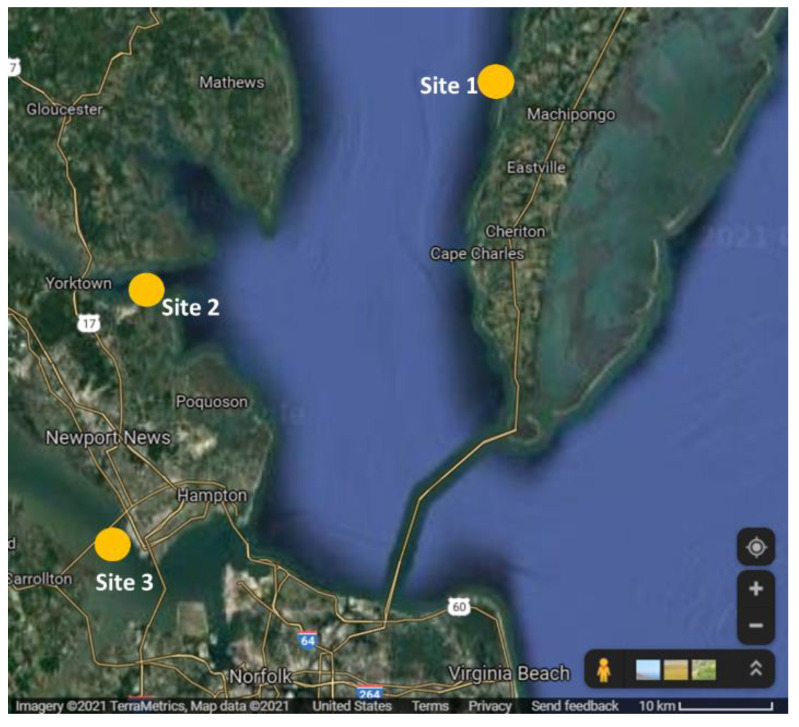
Sampling sites in the Chesapeake Bay in Virginia. Site 1: Eastern Shore; Site 2: York River; Site 3: James River.

**Figure 2 toxics-10-00029-f002:**
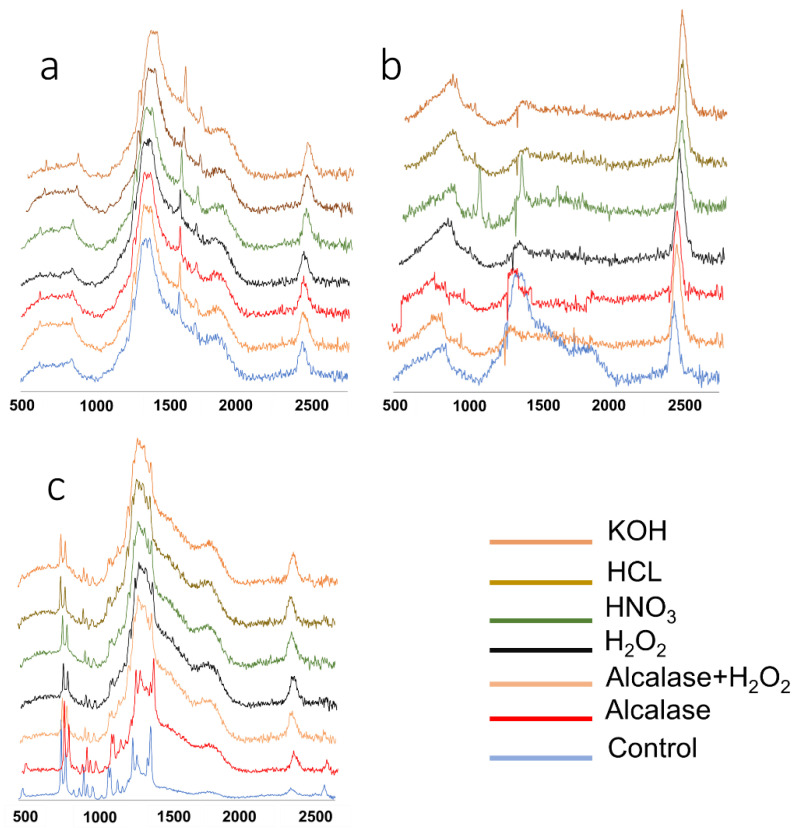
Raman spectra of (**a**) PET, (**b**) PS, and (**c**) PP exposed to different chemicals.

**Table 1 toxics-10-00029-t001:** Number of oysters collected from each site, and biometric information.

Site	Number of Oysters	Mean Weight of Soft Tissue (g)	Mean Shell Length (cm)
1	58	18.6 ± 2.1 ^a^	8.9 ± 1.6 ^a^
2	47	19.4 ± 2.1 ^a^	9.2 ± 1.2 ^a^
3	39	20 ± 3.4 ^a^	9.3 ± 1.18 ^a^

Values are mean ± sd. Values in the same column with different letter are significantly different (*p* < 0.05).

**Table 2 toxics-10-00029-t002:** The impact of digestion approaches on digestion efficiency, recovery rate, and morphological changes.

Digestion Approach	Standard Microplastics (Polyethylene) (300–355 μm)		Morphological Changes in Plastics		
	Digestion Efficiency	Recovery Rate	PP	PET	PS
Enzyme	57 ± 4 ^b^	38 ± 5 ^b^	-	-	-
Enzyme + H_2_O_2_ (30%)	62 ± 3 ^b^	35 ± 8 ^b^	-	-	-
H_2_O_2_ (30%)	100 ± 1.23 ^a^	92 ± 6 ^a^	-	-	-
HNO_3_ (69%)	100 ± 0 ^a^	0	Melted	Melted	Altered the color
HCl (5%)	48.1 ± 0.2 ^c^	42 ± 11 ^b^	Altered the color	Altered the color	Altered the color
KOH (10%)	100 ± 0.4 ^a^	96 ± 4 ^a^	Formed opaque color	Formed opaque color	Formed opaque color

Values are mean ± sd. Values in the same column with different letters are significantly different (*p* < 0.05). For Enzyme, Enzyme+H_2_O_2_ and H_2_O_2_ treatments (-) means no changes were observed.

**Table 3 toxics-10-00029-t003:** The number of microplastic types and total MPs per gram of tissue for each site.

Site	Fragment	Fiber	Bead
1	84 ± 18 ^a^	13 ± 6 ^a^	7 ± 4 ^a^
2	108 ± 23 ^a^	13 ± 6 ^a^	7 ± 5 ^a^
3	123 ± 32 ^a^	14 ± 6 ^a^	3 ± 4 ^a^

Values are mean ± sd. Values in the same column with different letter are significantly different (*p* < 0.05).

## Data Availability

The data are not publicly available due to protection of subjects’ privacy and confidentiality. The data presented in this study are available on request from the corresponding author.

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
