# Peer review of "Prevalence of Microplastics in the Eastern Oyster Crassostrea virginica in the Chesapeake Bay: The Impact of Different Digestion Methods on Microplastic Properties"

_toxics, 2022, doi:10.3390/toxics10010029_

Round 1

Reviewer 1 Report

First of all, thank you for your willingness to share your work with the scientific community. The manuscript entitled ‘Prevalence of microplastics in the eastern oyster Crassostrea virginica in the Chesapeake Bay: The impact of different digestion methods on microplastics properties’ describe different digestion approaches to digest oyster tissue without damaging microplastics. Authors also study MPs content in oysters collected in three sites in the Chesapeake Bay (Virginia).

The scope of the manuscript is interesting because of the scarce research in the oyster matrix, especially in the USA. Overall, the study is quite informative and well-presented. However, I have detected some drawbacks along with the manuscript. In the first one, there is no word about the sampling strategy and the sampling procedure itself. Secondly, if the first step has been to propose a digestion procedure, why has it not been used for the quantification of microplastics in oysters? This should be the natural procedure. Nevertheless, the authors used the chemical digestion method according to Li et al. and NOAA standard procedure for marine debris, why? Finally, it is well-known MPs are everywhere, so I miss a section or at least some lines about contamination controls. It is mandatory to prevent cross-contamination during lab procedures and to ensure the validity of the findings.

Other specific critical points must be clarified. I have included some suggestions below for ways to improve this manuscript.

Along the manuscript I suggest changing ml to mL, P < 0.05 to p < 0.05, and SD to sd. I also recommend revising subscripts (for example, in line 172 or line178)

Introduction section:

At this moment, it is available actual versions (2020) of Plastics the Facts and The estate of world Fisheries and aquaculture. I recommend actualizing these references and text.

In line 72 ‘…are only few studies…’ I suggest adding Bikker’s work (Bikker, J; Lawson, J; Wilson, S; Rochman, CM (2020) Microplastics and other anthropogenic particles in the surface waters of the Chesapeake Bay. Marine Pollution Bulletin, 156, 111257. Doi:10.1016/j.marpolbul.2020.111257)

Material and Methods section:

In lines 96-97 ‘..blue polyethylene microplastics…’ must be an error. The correct MP is polypropylene, isn’t it?

In subsection 2.1.1. units are lost. In line 98, what are the units of 60? 60ºC? In line 101, 24? 48? Please, you must revise these section units.

Approach 4 and approach 6 are identical. Please, revise them.

In approach 5, how long MP is digested?

In line 116, please, change 5 gram to 5 g

In subsection 2.1.2. What working range (cm-1) was used? What software was used?

In line145, I think there is a mistake with the volume flask, is it 5 mL?

In the last paragraph in subsection 2.2.2. I recommend adding units to 8.9, 9.2 and 9.3 cm? And 18.6, 19.4 and 20 g?

In section 2.3 I suggest including the procedure to measure MPs’ color and size.

Results section:

I suggest, in Table 3, including the standard deviation in Total MPs per g tissue.

Enzymatic digestion is a good option when you want to destroy organic matter while preserving microplastics. The problem with this digestion procedure is that the enzymes to be used must be carefully chosen depending on the matrix to be digested. For this reason, I suggest expanding the discussion of why this enzyme (alcalase) was chosen.

In lines 192-194 ‘…different digestion approaches on three plastics…’ what digestion approaches were used? Please, explain them in section 2.1.1.

In lines 216-222 the same information is available twice, I suggest rewriting these sentences. I also observe that in line 218 York River oysters’ present 129 MPs but Table 3 the amount is 128, please, check it.

In line 240 I recommend including a reference.

In line 252, please, change (van Cauwenberghe and Janssen, 2014) to the correct reference [28]

In line 253, ibidem, [29]

Author Response

First of all, thank you for your willingness to share your work with the scientific community. The manuscript entitled ‘Prevalence of microplastics in the eastern oyster Crassostrea virginica in the Chesapeake Bay: The impact of different digestion methods on microplastics properties’ describe different digestion approaches to digest oyster tissue without damaging microplastics. Authors also study MPs content in oysters collected in three sites in the Chesapeake Bay (Virginia).

The scope of the manuscript is interesting because of the scarce research in the oyster matrix, especially in the USA. Overall, the study is quite informative and well-presented. However, I have detected some drawbacks along with the manuscript. In the first one, there is no word about the sampling strategy and the sampling procedure itself.

Response: We added more information to the M&M regarding the sampling. Sampling was conducted with local seafood providers; however, we have not brought their names since the topic could be sensitive for aquaculture and oyster farmers.

Secondly, if the first step has been to propose a digestion procedure, why has it not been used for the quantification of microplastics in oysters? This should be the natural procedure. Nevertheless, the authors used the chemical digestion method according to Li et al. and NOAA standard procedure for marine debris, why?

Response: Among these six methods which we applied, Hydrogen peroxide did not change the quality of the standard plastics, and based on that we decided to use Hydrogen peroxide according to Li and NOAA. However, KOH is much more cheaper compared to Hydrogen peroxide.

Finally, it is well-known MPs are everywhere, so I miss a section or at least some lines about contamination controls. It is mandatory to prevent cross-contamination during lab procedures and to ensure the validity of the findings.

Response: Correct, this is very important and was one of our concerns. We filtered all the solutions, and washed our glass vessels several times. The information is available at the very beginning on the M&M as “During the entire experiment, to prevent any cross-contact, all the solvents were filtered using 0.45 um filter paper, and all the glassware were washed with 1 M nitric acid, rinsed three times with deionized water, and washed with 70% ethanol, and dried in an oven, and then covered with aluminum foil.”

Other specific critical points must be clarified. I have included some suggestions below for ways to improve this manuscript.

Along the manuscript I suggest changing ml to mL, P < 0.05 to p < 0.05, and SD to sd. I also recommend revising subscripts (for example, in line 172 or line178)

 Response: Revised.

Introduction section:

At this moment, it is available actual versions (2020) of Plastics the Facts and The estate of world Fisheries and aquaculture. I recommend actualizing these references and text.

 Response: Revised.

In line 72 ‘…are only few studies…’ I suggest adding Bikker’s work (Bikker, J; Lawson, J; Wilson, S; Rochman, CM (2020) Microplastics and other anthropogenic particles in the surface waters of the Chesapeake Bay. Marine Pollution Bulletin, 156, 111257. Doi:10.1016/j.marpolbul.2020.111257)

Response: Revised.  

Material and Methods section:

In lines 96-97 ‘..blue polyethylene microplastics…’ must be an error. The correct MP is polypropylene, isn’t it?

Response: Sorry for the typo, it is Polyethylene. I was revised.

In subsection 2.1.1. units are lost. In line 98, what are the units of 60? 60ºC? In line 101, 24? 48? Please, you must revise these section units.\

Response: Revised.  

Approach 4 and approach 6 are identical. Please, revise them. Response: Revised.  

In approach 5, how long MP is digested? Response: Revised.  

In line 116, please, change 5 gram to 5 g Response: Revised.  

In subsection 2.1.2. What working range (cm-1) was used? What software was used? Response: Revised.  

In line145, I think there is a mistake with the volume flask, is it 5 mL? Response: Revised.  

In the last paragraph in subsection 2.2.2. I recommend adding units to 8.9, 9.2 and 9.3 cm? And 18.6, 19.4 and 20 g? Response: Revised.  

In section 2.3 I suggest including the procedure to measure MPs’ color and size. Response: Revised.  

Results section:

I suggest, in Table 3, including the standard deviation in Total MPs per g tissue. Response: Revised.  

Enzymatic digestion is a good option when you want to destroy organic matter while preserving microplastics. The problem with this digestion procedure is that the enzymes to be used must be carefully chosen depending on the matrix to be digested. For this reason, I suggest expanding the discussion of why this enzyme (alcalase) was chosen. Response: Revised.  

In lines 192-194 ‘…different digestion approaches on three plastics…’ what digestion approaches were used? Please, explain them in section 2.1.1. Response: Revised.  

In lines 216-222 the same information is available twice, I suggest rewriting these sentences. I also observe that in line 218 York River oysters’ present 129 MPs but Table 3 the amount is 128, please, check it. Response: Revised.  

In line 240 I recommend including a reference. Response: Revised.  

In line 252, please, change (van Cauwenberghe and Janssen, 2014) to the correct reference [28] Response: Revised.  

In line 253, ibidem, [29] Response: Revised.  

Reviewer 2 Report

The manuscript entitled “Prevalence of microplastics in the eastern oyster Crassostrea virginica in the Chesapeake Bay: The impact of different digestion methods on microplastics properties” reports data of the presence of microplastics in oysters.

The article shows six different digestion methods to quantify microplastics in soft tissues. The samples were collected from Chesapeake Bay, the third-largest oyster producer in the U.S. The results illustrated that treatment with H2O2, KOH, and HNO3, had the highest digestion efficiency. The article also shows how digestion methods can change the chemical structure of plastics.

The paper is worthy of publication after minor changes.

Comments

It is not clear how many and which oysters were used for Experiment 1. Were all the 144 live oyster samples collected from three sites used in Experiment 2? Authors should specify data on samples for the Experiment 1.

Line 36: Regarding the detection of plastic polymers in marine organisms, authors should cite studies on crustaceans such as: Lo Brutto, S.; Iaciofano, D.; Lo Turco, V.; Potortì, A.G.; Rando, R.; Arizza, V.; Di Stefano, V. First Assessment of Plasticizers in Marine Coastal Litter-Feeder Fauna in the Mediterranean Sea. Toxics 2021, 9, 31. https://doi.org/10.3390/toxics9020031

References: please check the format, the number for each reference.

Author Response

The manuscript entitled “Prevalence of microplastics in the eastern oyster Crassostrea virginica in the Chesapeake Bay: The impact of different digestion methods on microplastics properties” reports data of the presence of microplastics in oysters.

The article shows six different digestion methods to quantify microplastics in soft tissues. The samples were collected from Chesapeake Bay, the third-largest oyster producer in the U.S. The results illustrated that treatment with H2O2, KOH, and HNO3, had the highest digestion efficiency. The article also shows how digestion methods can change the chemical structure of plastics.

The paper is worthy of publication after minor changes.

Comments

It is not clear how many and which oysters were used for Experiment 1. Were all the 144 live oyster samples collected from three sites used in Experiment 2? Authors should specify data on samples for the Experiment 1.

Response: Since this was optimization, we purchased oysters from the local seafood industry, and conducted the experiments in triplicates.

Line 36: Regarding the detection of plastic polymers in marine organisms, authors should cite studies on crustaceans such as: Lo Brutto, S.; Iaciofano, D.; Lo Turco, V.; Potortì, A.G.; Rando, R.; Arizza, V.; Di Stefano, V. First Assessment of Plasticizers in Marine Coastal Litter-Feeder Fauna in the Mediterranean Sea. Toxics 2021, 9, 31. https://doi.org/10.3390/toxics9020031

Response: Revised.  

References: please check the format, the number for each reference.

Response: Revised.